# Drug Resistance Analysis of Pancreatic Cancer Based on Universally Differentially Expressed Genes

**DOI:** 10.3390/ijms26093936

**Published:** 2025-04-22

**Authors:** Jie Xia, Linyong Zheng, Huarong Zhang, Qi Fan, Hui Liu, Ouxi Wang, Haidan Yan

**Affiliations:** 1School of Biology and Engineering, Guizhou Medical University, Guiyang 550025, China; vxiajie@163.com; 2Fujian Key Laboratory of Medical Bioinformatics, Department of Bioinformatics, School of Medical Technology and Engineering, Fujian Medical University, Fuzhou 350100, China; zhengly2025@163.com (L.Z.); hr_zhang@fjmu.edu.cn (H.Z.); funmoy@163.com (Q.F.); liuhui_fjmu@163.com (H.L.); fjmuqz@163.com (O.W.)

**Keywords:** pancreatic ductal adenocarcinoma, universally differentially expressed genes, drug resistance, immunity

## Abstract

The high heterogeneity between patients can complicate the diagnosis and treatment of pancreatic ductal adenocarcinoma (PDAC). Here, we explored the association of universally differentially expressed genes (UDEGs) with resistance to chemotherapy and immunotherapy in the context of pancreatic cancer. In this work, sixteen up-regulated and three down-regulated genes that were dysregulated in more than 85% of 102 paired and 5% of 521 unpaired PDAC samples were identified and defined as UDEGs. A single-cell level analysis further validated the high expression levels of the up-UDEGs and the low levels of the down-UDEGs in cancer-related ductal cells, which could represent the malignant changes seen in pancreatic cancer. Based on a drug sensitivity analysis, we found that *ANLN*, *GPRC5A* and *SERPINB5* are closely related to the resistance mechanism of PDAC, and their high expression predicted worse survival for PDAC patients. This suggests that targeting these genes could be a potential way to reduce drug resistance and improve survival. Based on the immune infiltration analysis, the abnormal expression of the UDEGs was found to be related to the formation of an immunosuppressive tumor microenvironment. In conclusion, these UDEGs are common features of PDAC and could be involved in the resistance of pancreatic cancer and might serve as novel drug targets to guide research into drug repurposing.

## 1. Introduction

Pancreatic ductal adenocarcinoma (PDAC) is considered one of the most aggressive malignant neoplasms, with a 12.8% five-year survival rate according to National Cancer Institute statistics. Complete surgery resection is the only possible cure for resectable PDAC, but most patients are already in an advanced stage at diagnosis [1]. Chemotherapy with/without radiotherapy has been a mainstay strategy for treating PDAC, but it only provides a modest improvement in survival [2]. Only 3% to 11% of patients have pathological complete responses after receiving chemotherapy with/without radiation therapy [3,4,5,6,7], which may be due to the high heterogeneity between patients and the ubiquitous drug resistance of PDAC. Immune checkpoint blockade (ICB) has changed the standard of care for multiple cancers and has brought hope to pancreatic cancer patients; however, almost all tentative trials have had little effect or failed before clinical application [8,9]. The most common reason for failure was the immunosuppressive tumor microenvironment (TME) resulting in low effector T cell infiltration, which is unfavorable for the immune response [10]. We suspect that there are some common features underlying the high heterogeneity of pancreatic cancer, which could be closely related to the mechanism underlying the ineffectiveness of current therapies. It is essential to identify and target these mechanisms to improve the early diagnosis rate and therapeutic efficiency.

Conventional differentially expressed gene (DEG) screening methods for bulk tissues only compare the average expression levels between disease and control groups, which cannot show the dysregulation status of each gene in a single disease sample or the frequency of differential expression of genes in all disease samples. The gold standard for individualized DEG screening is comparing the expression levels between paired tumor and adjacent normal samples from the same patient. However, the late diagnosis and difficulty of sampling pathologically normal samples in the complicated structure of the pancreas has led to few paired samples. To solve this problem, a robust individual DEG screening method, RankComp [11], has been developed to detect the dysregulation of genes in a single tumor sample relative to the normal background derived from normal samples from different studies. Thus, we can identify individual-level DEGs for unpaired tumor samples based on the aggregation of normal samples using this method. Here, we made full use of paired and unpaired PDAC samples to identify common molecular characteristics of PDAC patients to construct a robust early diagnosis model and determine the mechanisms underlying the resistance to current therapies.

In this study, using transcription profiles of paired and unpaired PDAC bulk samples, we identified individual-level DEGs in the PDAC samples and defined the genes that were dysregulated in at least 85% of the PDAC samples as universally differentially expressed genes (UDEGs). Bulk assays represent a population average, which masks the heterogeneity that exists at the single-cell level [12,13]. Thus, we analyzed the expression of the UDEGs at the single-cell level to explore their roles in the formation of tumors. Finally, we investigated the potential association between the UDEGs and overall drug resistance based on GDSC resistance data, as well as their correlation with responsiveness to immunotherapy using reported immunotherapy biomarkers.

## 2. Results

### 2.1. Identification of UDEGs in PDAC

The workflow for identifying UDEGs is shown in Figure 1A. In the 102 paired cancer and normal samples from GSE60646, GSE22780, GSE28735 and GSE15471, 180 genes were up-regulated and 32 genes were down-regulated in more than 85% of the tumor samples compared to the paired adjacent normal samples (Figure 1B). In the unpaired samples, based on 44,157,295 gene pairs that were stable in at least 90% of the 212 normal samples, DEGs were detected for each tumor sample using RankComp (FDR < 0.05). A total of 207 up-regulated and 153 down-regulated genes with dysregulated frequencies greater than 85% in 521 PDAC samples were detected. Finally, sixteen up-regulated and three down-regulated genes overlapped in the two DEG lists and were defined as UDEGs (Figure 1B–D).

### 2.2. Functions and Roles of UDEGs in PDAC

The P53 signaling pathway (*p* = 2.76 × 10^−3^) and cancer gene set from the CancerMine database (Hypergeometric test, *p* = 1.45 × 10^−3^) were enriched with UDEGs. Among them, *SFN*, *LAMC2*, *GPRC5A*, *ANLN*, *ESM1*, *SERPINB5* and *SLC6A14* have been reported to be closely related to the carcinogenesis and metastasis of PDAC (Appendix Table A1). *SFN* (Stratifin) is over-expressed and hypomethylated in cancer tissues compared with normal ducts and plays a role as an oncogene in pancreatic cancer [14,15]. *LAMC2* (laminin subunit gamma 2) promotes cell proliferation, invasion and migration in multiple cancers, including pancreatic cancer [16,17,18]. Knockout of *GPRC5A* (G protein-coupled receptor family C, group 5, member A) reduced the proliferation and migration ability of pancreatic cancer cell lines and suppressed the resistance of pancreatic cancer cell lines to the chemotherapy drugs gemcitabine, oxaliplatin and fluorouracil [19]. The up-regulation of anillin (*ANLN*) promotes the progression and metastasis of PDAC [20,21,22]. *ESM1* encodes a secreted protein that is mainly expressed in endothelial cells; it regulates the proliferation, migration, invasion and drug resistance of tumor cells, thereby promoting tumor progression and metastasis and playing an important role in the pathogenesis of pancreatic cancer [23]. Several reports have shown that the over-expression of *SERPINB5* (serine protease inhibitor B5) can promote PDAC metastasis [24], and the marked up-regulation of *SLC6A14* (solute carrier, family 6, member 14) in pancreatic cancer leads to worse survival [25,26,27]. Overall, these UDEGs are important in the carcinogenesis, invasion and metastasis of PDAC.

To further analyze the biological functions of the UDEGs, we generated a PPI network for the 19 UDEGs and found 155 proteins that interact with them (Figure 2A). Several carcinogenesis- and metastasis-related pathways, such as the cell cycle, focal adhesion, ErbB signaling and PI3K-Akt signaling pathways, are enriched with these 174 genes (Figure 2B). According to the Reactome pathway analysis, immune system, signal transduction, cell cycle, gene expression, disease, programmed cell death and extracellular matrix pathways were hierarchically enriched with the UDEGs (Figure 2C). These results indicate that in addition to cell proliferation and adhesion, changes in the immune system are also important for the carcinogenesis and development of pancreatic cancer.

### 2.3. Characteristics of UDEGs at Single-Cell Level

Because a bulk cancer tissue sample is a mixture of different types of cells [12,13], we explored whether the UDEGs are specifically dysregulated in cancer cells at the single-cell level. Here, we analyzed scRNA-seq profiles from CRA001160, which contains 24 PDAC and 11 control samples. From the tumor samples, 42,066 cells were acquired after quality control, which were separated into 39 clusters (Figure 3A), classified as acinar cells (cluster 20), B cells (clusters 7, 22 and 25), macrophages (clusters 1, 15, 32 and 37), endocrine cells (cluster 31), neuroendocrine cells (cluster 36), fibroblasts (clusters 5, 6, 23, 24, 33 and 38), endothelial cells (clusters 0 and 11), monocytes, stellate cells (clusters 2 and 4) and T cells. In particular, clusters 9, 10, 12, 13, 14, 16, 18, 19, 21, 26, 27, 28, 30, 34 and 35 not only expressed ductal cell markers (*KRT19*, *MMP7*, *TSPAN8*, *SOX9* and *LCN2*), but also expressed cancer cell markers (*MUC1* and *PROM1*). Thus, we defined these clusters as cancer-related ductal cells, and cluster 3, which only expressed ductal cell markers, was defined as ductal cells with normal characteristics.

From the 11 control samples, 15,553 cells passed the quality control and were classified into 27 clusters (Figure 3B), including acinar cells (clusters 7 and 15), ductal cells (clusters 0, 2, 4, 5, 6, 9, 11, 14, 16, 19 and 26), endocrine cells (cluster 20), endothelial cells (clusters 1, 8, 10, 17, 18 and 21), fibroblasts (cluster 3), stellate cells (cluster 12) and a small number of immune cells (T cells (clusters 22 and 24), monocytes (cluster 25), B cells (cluster 23) and macrophages (cluster 13)).

The major distinctions between the tumor and control samples were the large amount (27%) of cancer-related ductal cells and large proportion of immune cells in the tumor samples (25.29% in tumor samples compared with 4.15% in control samples) (Figure 3C,D). Although these cells are from merging different samples, the result still reflects the appearance of malignant lesions and an increase in inflammation in pancreatic cancer. In particular, up-UDEGs were specifically and highly expressed in cancer-related ductal cells while down-UDEGs had low expression (Figure 3E,F). Conversely, the down-UDEGs were expressed at high levels in the control samples, especially in some cell types, such as ductal cells, endothelial cells and fibroblasts. The KEGG enrichment analysis of specific highly expressed genes in each cell type found that most cells exhibit similar functions in both tumor and normal samples, such as T cells, macrophages or endocrine cells. However, multiple cancer-related pathways, such as necroptosis, pancreatic cancer, chemical carcinogenesis and reactive oxygen pathways, and metabolic pathways associated with proliferation, glutathione metabolism and glycolysis/gluconeogenesis (Table 1) were enriched with the DEGs from cancer-related ductal cells, similar to the enrichment in bulk tissues. This result indicates that cancer-related ductal cells contribute most to the markedly high expression levels of up-UDEGs in the tumor tissues compared to the control samples and to the cancer-related functions in PDAC tissues.

### 2.4. UDEGs Involved in Resistance to Anticancer Drugs

Drug resistance is an important cause of poor chemotherapy responses in pancreatic cancer. We posited that the UDEGs we identified are involved in the universal drug resistance of PDAC patients. We downloaded the expression profiles of 32 untreated pancreatic cell lines and IC50 values of the 448 anticancer drugs used to treat the 32 pancreas cancer cell lines from the GDSC database. The 412 drugs that were used to treat more than half of these cell lines were retained. The mRNA expression profiles of cell lines were measured using the Affymetrix Human Genome U219 Array, which only contains nine of the nineteen UDEGs. Therefore, we only analyzed the relationship between these nine genes and drug resistance in our study. Based on the Spearman correlation analysis of the expression of the nine UDEGs and the IC50 for pancreatic cancer cell lines, 371 drug–gene relationships were found to be significantly correlated (Figure 4A, Appendix Table A2).

Interestingly, most of the relationships between drug resistance and the expression levels of the up-UDEGs *ANLN*, *GPRC5A* and *SERPINB5* were positive (Figure 4B), indicating that these three genes are closely related to the resistance mechanisms of pancreatic cancer. Moreover, we found that high expression levels of *ANLN*, *GPRC5A* and *SERPINB5* were significantly associated with poor OS (overall survival) for TCGA PDAC patients (Figure 4C; *ANLN*: *p* = 0.034; *GPRC5A*: *p* = 0.045; *SERPINB5*: *p* = 0.028; log-rank test). Combining the above results, the universal drug resistance of patients with high expression of *ANLN*, *GPRC5A* and *SERPINB5* might be one of the causes of poor survival. Thus, targeting these genes could be an effective way to suppress drug resistance and improve survival.

Among the significant correlations between *LRP8* expression and the IC50s of anticancer compounds, 43/50 (86%) were negative (Figure 4B), indicating that most anticancer drugs exhibit higher efficacy when *LRP8* is highly expressed; these drugs included tamoxifen (Rho = −0.49, *p* = 4.83 × 10^−3^), rapamycin (Rho = −0.51, *p* = 3.58 × 10^−3^) and afuresertib (Rho = −0.50, *p* = 5.39 × 10^−3^). Tamoxifen is widely used in the treatment of estrogen receptor-positive breast cancer and has been found to inhibit the myofibroblastic differentiation of pancreatic stellate cells (PSCs) and cell invasion into the tumor microenvironment [28], but it has not been used in the treatment of pancreatic cancer. Rapamycin is a lipophilic macrolide antibiotic that was originally developed as a fungicide and an immunosuppressant. Combined with cisplatin, it can affect the *PI3K/AKT/mTOR* signal transduction pathway, which leads to markedly increased cell apoptosis, indicating that rapamycin can mediate the sensitivity of pancreatic cancer cells to cisplatin [29]. Afuresertib is a novel inhibitor of the serine/threonine kinase *AKT* and has shown clinical efficacy as a monotherapy against hematological malignancies and could be used in combination with standard therapies for multiple myeloma [30]. In summary, our results identified some potential novel drugs for pancreatic cancer therapy that deserve further clinical research.

We also explored drug repurposing based on the UDEGs by generating a DGI network for the UDEGs and their PPI partners (Figure 4D). Some anticancer drugs (amatuximab, AZD-4877, ispinesib, letrozole, cergutuzumab amunaleukin, filanesib and nutlin-3), type II diabetes drugs (omega-3-carboxylic acids and ruboxistaurin) and drugs used to stimulate gastric and pancreatic secretions in hyperlipoproteinemia (levocarnitine) were found to interact with the UDEGs. Moreover, some novel drugs that have not been used to treat PDAC-related diseases were found to interact with the UDEGs, such as a drug for ocular hypertension (bimatoprost) and the antiepileptic drug losigamone. These results not only indicate that the UDEGs could be drug targets for pancreatic cancer but also provide new insights for drug repurposing to treat pancreatic cancer.

### 2.5. Roles of UDEGs in Response to Immunotherapy for PDAC Patients

The challenge in treating pancreatic cancer lies in its resistance to both chemotherapy and radiotherapy, as well as its unresponsiveness to the increasingly popular immunotherapy approach. Thus, we explored the roles of the UDEGs in the immune response to immunotherapy in PDAC patients.

A commonly used biomarker of immunotherapy, TMB, is universally low in PDAC patients, with an average of 0.73 mutations/megabase. For comparison, cancers with a high response rate to immunotherapy, such as melanoma and squamous cell lung cancer, have TMBs of 14.77 and 5.82 mutations/megabase, respectively (Appendix Figure A1A). Another marker, MSI-high, has been reported as a potential predictor of the effectiveness of immunotherapy in PDAC patients [31]. The TCGA mononucleotide and dinucleotide marker panel analysis found 24 PDAC patients with an indeterminate status, 115 with an MSS (microsatellite stable) status, 8 with an MSI-L (low MSI) status, and no patients with an MSI-H (high MSI) status (Appendix Figure A1B). Thus, it is difficult to apply TMB and MSI-H to measure immunotherapy responsiveness in PDAC.

The TME of PDAC is changeable, and modulation of the cell state can strongly influence drug responses, with state-specific vulnerabilities demonstrated in vivo and ex vivo [32]. Thus, we identified the proportions of 22 tumor-infiltrating lymphocytes in the TME of the TCGA PDAC patients using the CIBERSORT deconvolution algorithm. Consistent with previous studies, pro-tumor immune cells such as M0 and M2 macrophages and CD^4+^ memory resting cells were relatively abundant, but effector T cells such as CD^4+^ T cells and CD^8+^ T cells were relatively rare (Figure 5A), indicating a strong pro-tumor property and an immunosuppressive TME in PDAC. The results showed that most up-regulated UDEGs were significantly negatively correlated with the proportion of TME components associated with the response to ICB, including CD^8+^ T cells, naive B cells and activated dendritic cells, and significantly positively correlated with the proportion of TME components associated with resistance to ICB, including M0 macrophages, resting dendritic cells and Tregs (Figure 5B). Moreover, the down-UDEGs *CYB5A* and *EPHX1* were significantly positively correlated with the proportion of CD^8+^ T cells, which have been reported to be associated with ICB responsiveness (Figure 5B). These results suggest that high expression of the up-regulated UDEGs and low expression of the down-regulated UDEGs are related to the formation of an immunosuppressive TME and might predict an unfavorable response to immunotherapy.

## 3. Discussion

The high heterogeneity between patients and the ubiquitous drug resistance are the main reasons for the poor efficacy of the current therapies for pancreatic cancer. Understanding the common molecular characteristics is essential to understanding the mechanisms of this universal therapeutic resistance and might be the key to overcoming this resistance. In this work, sixteen up-regulated and three down-regulated UDEGs were identified using paired cancer and normal samples and unpaired cancer bulk tissues. The single-cell analysis validated the high expression levels of the up-UDEGs and low levels of the down-UDEGs in cancer-related ductal cells compared to those in the control samples, which could represent the malignant changes in pancreatic cancer.

Among the nineteen UDEGs, other than the seven UDEGs reported to be carcinogenic and prometastatic genes involved in pancreatic cancer, the other UDEGs, such as *ESM1*, *PCDH7*, *CORO2A*, *CEACAM5*, *GALNT5* and *TMPRSS4*, are also involved in other cancers. *ESM1* (endothelial cell-specific molecule 1) can promote cancer progression and metastasis through the regulation of tumor cell proliferation, migration, invasion and drug resistance. *PCDH7* (protocadherin 7) was reported to be related to cell growth, development and progression in prostate cancer [33], cervical cancer [34] and breast cancer [35]. *CORO2A* (coronin 2A) plays a critical role in cell migration and proliferation in breast cancer [36]. *CEACAM5* (carcinoembryonic antigen-related cell adhesion molecule 5, also known as *CEA*) was reported to be a driver gene in colorectal cancer [37,38] and is related to tumor differentiation, invasion and metastasis [39]. *GALNT5* (GalNAc transferase 5) mediates the carcinogenesis and progression of cholangiocarcinoma [40] and gastric cancer [41]. *LRP8* is over-expressed in multiple cancers and is a potential therapeutic target [42,43,44,45]. *TMPRSS4* is an important mediator of cell migration, invasion, epithelial–mesenchymal transition and metastasis in colon cancer cells, and increased *TMPRSS4* expression correlated with colorectal cancer stage progression [46].

Based on these UDEGs, we performed a drug resistance and immune infiltration analysis and investigated the correlation between the expression of the UDEGs and the response to treatments for PDAC. These UDEGs were found to be closely related to the ubiquitous resistance of pancreatic cancer. For example, the higher expression of *ANLN* and *GPRC5A* was correlated with higher resistance to gemcitabine in pancreatic cell lines (*ANLN*: R = 0.43, *p* = 1.5 × 10^−2^; *GPRCA5*: R = 0.53, *p* = 2.3 × 10^−3^; Spearman correlation, Appendix Figure A2A). In contrast, in the TCGA PDAC cohorts, the non-responsive patients tended to express higher levels of *ANLN* and *GPRC5A* than the responsive patients (Appendix Figure A2B). Thus, PDAC patients with higher levels of *ANLN* and *GPRC5A* expression are more likely to be resistant to gemcitabine at the cell and tissue levels. Moreover, previous studies have shown that a down-regulation of *ANLN* can enhance the sensitivity of pancreatic cancer cells to gemcitabine [47] and inhibit doxorubicin resistance in human breast cancer cells [48]. Another study proved that knocking out *GPRC5A* suppressed the resistance to gemcitabine in a pancreatic cancer cell line (MIA PaCa-2) [19], and its over-expression enhanced the resistance of ovarian cancer cells [49]. *SERPINB5* over-expression has been shown to promote metastasis in pancreatic ductal adenocarcinoma, and knockdown of *SERPINB5* reduced the primary tumor weight and significantly decreased the amount of metastasis [24]. However, it has not been validated to be related to drug resistance and needs further validation through experiments. These results indicate that the UDEGs could be potential drug targets for pancreatic cancer. Thus, we also performed a DGI network analysis and found that some drugs used in other diseases interact with the UDEGs or their PPI partners that play important roles in the treatment of pancreatic cancer. These drugs are approved by the FDA and have passed preclinical testing, which not only ensures their safety but also greatly shortens the time for drug development and reduces the risk of failure.

In conclusion, we identified 19 universally differentially expressed genes (16 up-regulated and 3 down-regulated) in PDAC through a combination of bulk and single-cell analyses. These UDEGs are closely related to the drug resistance of PDAC, providing several potential targets for repurposing anticancer drugs to treat PDAC.

## 4. Materials and Methods

### 4.1. The Sources and Preprocessing of Expression Profiles

All the data used in our study were downloaded from public databases. Transcription profiles of PDAC and adjacent normal samples from PDAC patients and normal samples from pancreatic disease-free people came from the GEO (ncbi.nlm.nih.gov/geo, accessed on 13 March 2022), TCGA (cancergenome.nih.gov, accessed on 23 March 2022), GTEx (The Genotype-Tissue Expression, gtexportal.org, accessed on 13 March 2022) and ArrayExpress (ebi.ac.uk/arrayexpress, accessed on 13 March 2022) databases (Table 2).

For the data obtained using the Affymetrix platform, we applied the robust multi-array average algorithm for background adjustment without inner-sample normalization. Every probe was mapped to an Entrez gene ID according to the corresponding platform annotation file. The probes that mapped to multiple genes were discarded. If multiple probes were mapped to the same gene, the expression value of this gene was defined as the arithmetic mean of the values of the mapped probes. For the data sets obtained using the Illumina and Agilent platforms, we directly downloaded the processed expression data. For the profiles from the TCGA database generated using RNA-seq, the level 3 data were directly downloaded. Then, the Ensemble gene IDs were mapped to the Entrez gene IDs of protein-coding genes and used for further analysis. In this work, the PDAC and adjacent normal samples from the same patient were called paired samples; otherwise, they were referred to as unpaired samples.

### 4.2. Identification of Universally Dysregulated Genes in PDAC

We screened UDEGs using both paired and unpaired PDAC samples from public databases. In the paired samples, if the expression of a gene in the tumor sample was higher than that in the paired adjacent normal sample, it was defined as up-regulated and vice versa. For the unpaired samples, we screened dysregulated genes in individual samples by applying the RankComp method [11]. First, the normal background was derived from stable gene pairs, namely those gene pairs in which the rank of one gene is higher than or lower (G_i_ < G_j_ or G_i_ > G_j_) than that of the other gene in more than 90% of the normal samples from different sources. Next, reverse gene pairs were defined for each disease sample as gene pairs with reversed rank ordering in comparison with their ordering in normal samples (G_i_ > G_j_ → G_i_ < G_j_ or G_i_ < G_j_ → G_i_ > G_j_). Afterwards, Fisher’s exact test was used to test the null hypothesis that the numbers of reverse gene pairs supporting the up-regulation and down-regulation of a given gene (G_i_) in a given disease sample (k) are equal. Based on the ratios of gene pairs with different ordering patterns in the normal and disease samples, we determined whether G_i_ is up-regulated (if the ratio in the disease sample is greater than in the normal samples) or down-regulated (if the ratio is smaller) or has stable expression (if the ratios are equal). A gene was determined to be a differentially expressed gene only if it remained significant after excluding the down-regulated (or up-regulated) partner genes from the reverse gene pairs supporting its up-regulation (or down-regulation) designation. Finally, the dysregulation frequency of each gene in the two approaches was calculated, and the overlapping genes with more than 85% frequency in both DEG lists were retained as UDEGs for the subsequent analyses.

### 4.3. Protein–Protein Interaction Network

A protein–protein interaction (PPI) network was created using PPIs from the STRING database [50]. Each PPI is annotated with a confidence score, which represents how likely STRING judges an interaction to be true given the available evidence. The scores range from 0 to 1, with 1 representing the highest possible confidence. To ensure the high confidence of the PPIs used to generate our network, we set the interaction score threshold to 0.99.

### 4.4. Prognostic Efficiency and Functions of UDEGs

A survival analysis was performed on the TCGA PDAC cohort using the R package “survival” (version: R 4.0.2)to explore the prognostic efficacy of UDEGs. The CancerMine [51] database was searched to explore the reported roles of the UDEGs in multiple cancers. To analyze the functions of the UDEGs, a KEGG pathway analysis was conducted, and REACTOME [52] (reactome.org, accessed on 20 May 2022) was used to systematically display the hierarchy of the pathway enrichment in this study.

### 4.5. Processing of scRNA-Seq Data

The raw scRNA-seq data were downloaded from the GSA database (Genome Sequence Archive, ngdc.cncb.ac.cn/gsa, accessed on 21 April 2022, CRA001160 [53]). The “Seurat” [54] (v2.3.0) R package was used for quality control and downstream single-cell analysis. For quality control, we set the following criteria: <200 genes/cell, <3 cells/gene and >10% mitochondrial genes/cell. The tumor and control samples were merged and scaled.

### 4.6. Cluster Identification and Cell Type Assignment

The top 2000 variable genes were used for clustering. T-SNE reduction was applied to display the clustering results. The cluster-specific marker genes were identified by running the “FindAllMarkers” function of the “Seurat” package. We used some well-known markers from the CellMarker database [55] and the literature to identify specific cell types; for example, we used *CPA1*, *PRSS1*, *CTRB1*, *CTRB2* and *REG1B* to identify acinar cells; *MS4A1* and *CD27* to identify B cells; *AIF1* and *CD68* to identify macrophages; *PCSK1N*, *CHGB*, *CHGA*, *INS* and *SLC30A8* to identify endocrine cells; *KRT19*, *MMP7*, *TSPAN8*, *SOX9* and *LCN2* to identify ductal cells; *MUC1* and *PROM1* to identify cancer cells; *LUM*, *DCN* and *LRP1* to identify fibroblasts; *MNDA* and *ITGB2* to identify monocytes; *RGS5*, *ACTA2*, *PDGFRB* and *ADIRF* to identify stellate cells; and *CD2*, *CD3D* and *CD3E* to identify T cells. Using Seurat’s *FindAllMarkers* function, we identified genes that were highly expressed in acinar cells, duct cells, macrophages, endocrine cells, fibroblasts, endothelial cells, monocytes, stellate cells and T cells in pancreatic cancer and normal tissues. Then, KEGG functional enrichment analysis was performed on each gene list (*p* <= 0.05). The functional conformance of common cell types between the pancreatic and normal tissue samples was determined using the following formula:$$Consistence=(\frac{con}{n}+\frac{con}{t})/2$$

For a given cell type, n is the number of pathways enriched with the DEGs of this cell in the normal tissues, t is the number of pathways enriched with the DEGs of this cell in the cancer tissues and con is the number of common pathways in both the cancer and normal tissues.

### 4.7. Drug Resistance

The processed expression profiles and sensitivity data of 32 pancreatic cell lines were downloaded from the GDSC database (Genomics of Drug Sensitivity in Cancer [56], cancerrxgene.org, accessed on 21 June 2022). The sensitivity of a cell line to a drug is represented by the IC50 value. For each drug–gene pair, we calculated the Spearman correlation coefficient between the expression values of the gene and the IC50 values of the drug across different cell lines. Only the drugs that were effective against at least half of the 32 cell lines and had drug–gene correlations with a *p*-value < 0.05 were retained.

### 4.8. Drug–Gene Interaction Network

We explored the potential of the UDEGs to be drug targets by constructing a drug–gene interaction (DGI) network using information from DGIdb [57] (dgidb.genome.wustl.edu, accessed on 23 June 2022). We select drug–gene interactions with scores greater than 5 as candidate interactions and obtained the drugs’ FDA approval information, mechanism and function from the DrugBank database (go.drugbank.com/drugs, accessed on 23 June 2022).

### 4.9. TMB and MSI Information

The “TCGAmutations” [58] R package was used to calculate the tumor mutation burden (TMB) of the TCGA PDAC, SKCM (skin cutaneous melanoma) and LUSC (lung squamous cell carcinoma) patients. The “TCGAbiolinks” R package was used to download the microsatellite instability (MSI) information of the TCGA patients.

### 4.10. Immune Cell Proportion of TCGA PDAC Patients

CIBERSORT [59] was applied to obtain the proportions of 22 immune cell types, including macrophages (M0, M1 and M2), CD^8+^ T cells, naive CD^4+^ T cells, CD^4+^ memory resting T cells, CD^4+^ memory activated T cells, monocytes, resting dendritic cells, neutrophils, resting mast cells, naive B cells, eosinophils, memory B cells, resting NK cells, activated NK cells, follicular helper T cells, regulatory T cells (Tregs), gamma delta T cells, activated dendritic cells, plasma cells and activated mast cells. The correlations between the expression of the UDEGs and cell proportions were calculated using Spearman correlation.

### 4.11. Statistical Analysis

All statistical analyses were performed using R 4.0.2.

## Figures and Tables

**Figure 1 ijms-26-03936-f001:**
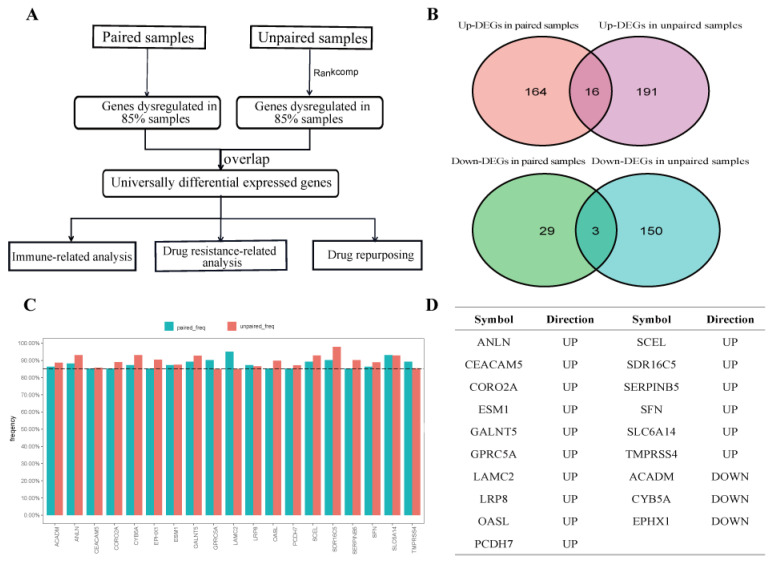
Identification of UDEGs in PDAC. (**A**) Workflow for identifying UDEGs in PDAC. (**B**) Overlapping genes in two DEG lists were defined as UDEGs. (**C**) Dysregulation frequencies of nineteen UDEGs. (**D**) Brief information on UDEGs.

**Figure 2 ijms-26-03936-f002:**
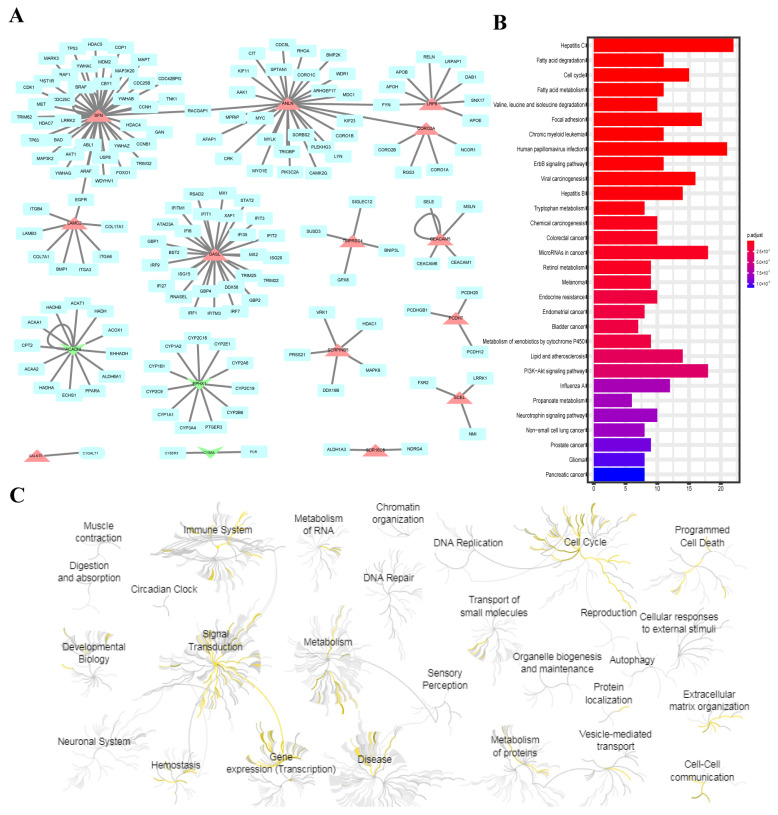
Functional analysis of UDEGs. (**A**) UDEGs and their direct neighbors in PPI network. Light blue represents neighbor genes, red represents upregulated genes, and green represents downregulated genes. (**B**) KEGG enrichment analysis of UDEGs and their direct neighbor genes. (**C**) Pathways with different functions that are hierarchically enriched with UDEGs according to Reactome.

**Figure 3 ijms-26-03936-f003:**
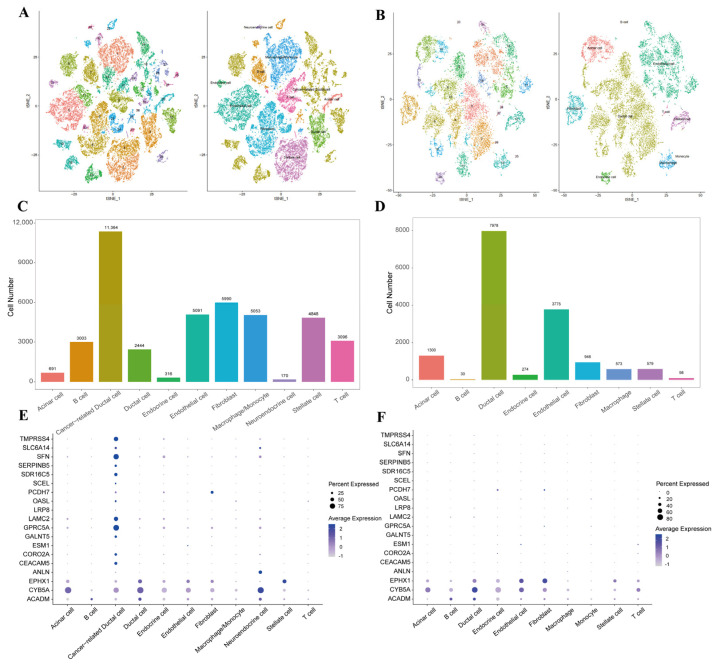
Single-cell analysis of 24 PDAC and 11 control samples. t-SNE plot of clustering and cell-type annotation of tumor (**A**) and normal (**B**) samples. Cell numbers for each cell type in tumor (**C**) and normal (**D**) samples. Expression of UDEGs in different cell types in tumor (**E**) and normal (**F**) samples.

**Figure 4 ijms-26-03936-f004:**
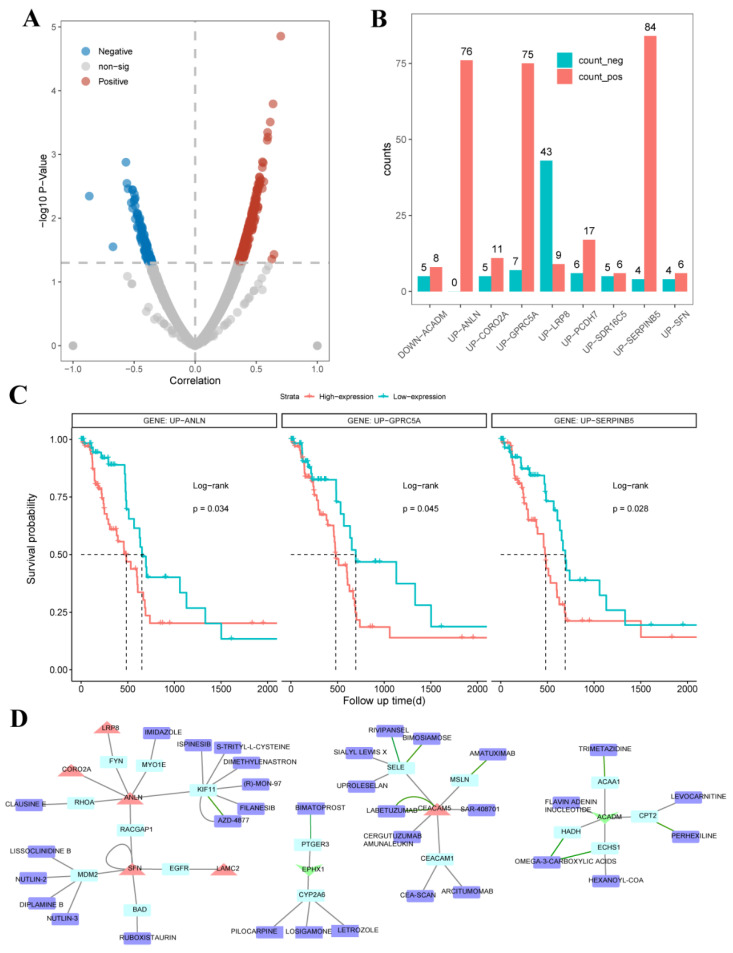
Correlation analysis between expression of UDEGs and resistance to anticancer drugs. (**A**) Overview of correlations between IC50 of drug in pancreatic cells and UDEG expression. (**B**) Overview of significant correlations between drug resistance and expression of UDEGs. (**C**) Survival analysis of *ANLN*, *GPRC5A* and *SERPINB5* in TCGA PDAC patients. (**D**) DGI network of UDEGs and their PPI partners. Light blue represents neighbor genes, red represents upregulated genes, green represents downregulated genes, and purple represents drugs.

**Figure 5 ijms-26-03936-f005:**
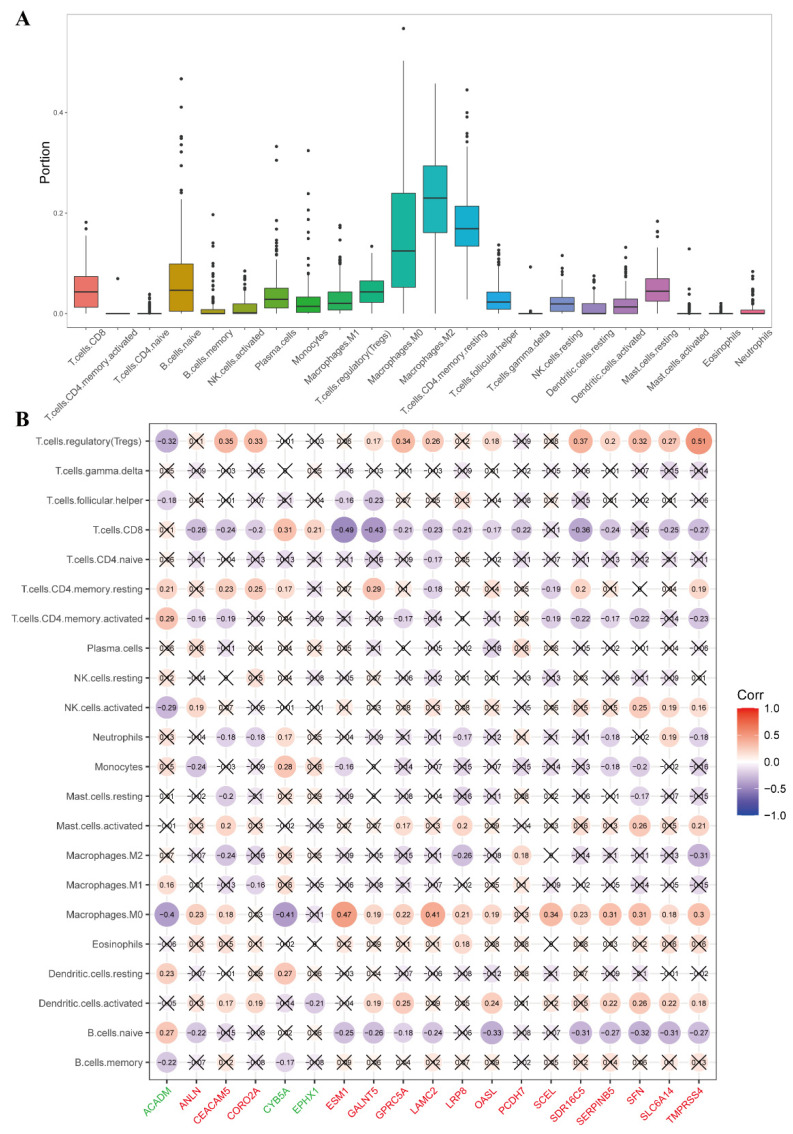
Roles of UDEGs in response of PDAC patients to immunotherapy. (**A**). Overview of immune cell proportions in TCGA PDAC patients. (**B**). Correlation between UDEGs and immune cell proportions. “×” represents a non-significant correlation. Genes labeled in green are down-regulated, and those in red are up-regulated.

**Table 1 ijms-26-03936-t001:** Functional enrichment analysis of different cell types in single-cell analysis of pancreatic cancer.

Cell Type	Number of Pathways			Consistance
	Normal	Tumor	Common	
Acinar cells	5	6	3	55.00%
Cancer-related ductal cells	-	29	-	-
Ductal cells	35	30	22	68.10%
Endocrine cells	27	29	22	78.67%
Endothelial cells	94	82	64	73.07%
Fibroblasts	22	29	16	63.95%
Macrophages	70	74	64	88.96%
Stellate cells	39	28	23	70.56%
T cells	30	32	28	90.42%

**Table 2 ijms-26-03936-t002:** Overview of samples used in this study.

GEO Accession	Platform	Sample Size		
		PDAC	PDAC_Adjacent	Normal
Unpaired Samples				
GSE62452	GPL6244	69	61	0
E-MTAB-1791	A-MEXP-2271	268	74	41
GSE91035	GPL22763	25	0	8
GSE62165	GPL13667	118	13	0
GSE56560	GPL5175	28	4	3
GSE71989	GPL570	13	0	8
Total		521	212	
Paired Samples				-
GSE60646	GPL5175	10	10	-
GSE22780	GPL570	8	8	-
GSE28735	GPL6244	45	45	-
GSE15471	GPL570	39	39	-
Total		102	102	-

PDAC: pancreatic ductal adenocarcinoma.

## Data Availability

The original data presented in the study are openly available in Gene Expression Omnibus (GEO, ncbi.nlm.nih.gov/geo), The Cancer Genome Atlas (TCGA, cancergenome.nih.gov), The Genotype-Tissue Expression (GTEX, gtexportal.org), ArrayExpress (ebi.ac.uk/arrayexpress), GSA database (Genome Sequence Archive, ngdc.cncb.ac.cn/gsa), Genomics of Drug Sensitivity in Cancer (GDSC, cancerrxgene.org), DGIdb (dgidb.genome.wustl.edu), and DrugBank database (go.drugbank.com/drugs).

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
