# Peer review of "Drug Resistance Analysis of Pancreatic Cancer Based on Universally Differentially Expressed Genes"

_ijms, 2025, doi:10.3390/ijms26093936_

Round 1
Reviewer 1 Report
Comments and Suggestions for Authors
In this manuscript, Xia et al aimed to explore the association of universally differentially expressed genes with prevalent resistance of chemotherapy and immunotherapy in pancreatic ductal adenocarcinoma. They analyzed paired and unpaired PDAC samples from publicly available databases and they concluded to 16 up-regulated and 3 down-regulated genes. They then validated the expression levels of these genes in ductal cells specifically. Finally, drug sensitivity analysis identified 3 genes to be associated with PDAC treatment resistance and worst survival. This topic is of high importance for a very aggressive cancer type, and the manuscript is of high potential. However, it is poorly written, convoluted, and difficult to follow.
1) It requires extensive English editing to be considered for publication.
2) No patient characteristics were reported for the datasets that were included in the study. The applicability and reproducibility of the results is contingent upon information on patient demographics, TNM classification, prior receipt of therapeutics etc.
3) There are several inconsistencies throughout the manuscript. For example, the first paragraph of the results section mentions that there are 153 up-regulated and 207 down-regulated genes in unpaired samples, but Figure 1B shows the opposite pattern. Please proofread.
4) The results section includes extensive discussion of existing literature. I would suggest focusing on explaining clearly the results of the study in this section and then compare and contrast with existing literature in the Discussion section.
5) What does "PDAC" stand for? Please make sure that all abbreviations are defined.
Comments on the Quality of English LanguageThe manuscript requires extensive English editing to be considered for publication.
Author Response
Comment 1: It requires extensive English editing to be considered for publication.
Response 1: We sincerely appreciate your feedback on the English in our manuscript. We fully understand the importance of clear and error-free language for publication. We have already utilized the manuscript polishing service provided by the MDPI and provided a certificate of English language editing along with the revised manuscript to demonstrate the thoroughness of this process. We believe this service will effectively address the language-related issues in our paper. The professional editors have carefully reviewed and refined our text to improve its grammar, syntax, and overall readability.
We are confident that after this round of editing, our manuscript will meet the language standards required for publication. If there are still any areas that you feel need further improvement, please let us know, and we will make every effort to address them.
Comment 2: No patient characteristics were reported for the datasets that were included in the study. The applicability and reproducibility of the results is contingent upon information on patient demographics, TNM classification, prior receipt of therapeutics etc.
Response 2: Thank you for your thoughtful review and the concerns you raised regarding the lack of reported patient characteristics in our study. We truly value your input as it contributes to the overall quality of our research.
However, we would like to clarify the situation regarding the data. The datasets we utilized in this study were sourced from publicly available repositories, such as GEO and TCGA. These repositories are widely recognized for their comprehensive nature and high - quality data management. While we understand the importance of patient - specific information like demographics, TNM classification, and prior therapeutic history, in the context of our analysis, we focused on a more generalizable approach to capture a broad spectrum of molecular patterns related to PDAC relying on large - scale, publicly - available datasets.
We also believe that the reproducibility of our results is ensured through the detailed description of our data processing and analysis methods in the manuscript. Other researchers can access the same datasets and replicate our analysis using the techniques we have outlined. While patient - specific information is important, for this study, our approach can still offer valuable insights. We plan to explore this aspect further in future research.
Comment 3: There are several inconsistencies throughout the manuscript. For example, the first paragraph of the results section mentions that there are 153 up-regulated and 207 down-regulated genes in unpaired samples, but Figure 1B shows the opposite pattern. Please proofread.
Response 3: We are extremely sorry for the inconsistencies in the manuscript, especially the discrepancy between the text in the Results section and Figure 1B. We will carefully proofread the entire manuscript to identify and correct all such inconsistencies. For the specific issue of the gene regulation numbers, we will double - check our data analysis process. If there was an error in data entry or analysis, we will correct it and update both the text and the figure accordingly.
Comment 4: The results section includes extensive discussion of existing literature. I would suggest focusing on explaining clearly the results of the study in this section and then compare and contrast with existing literature in the Discussion section.
Response 4: We agree with your suggestion to separate the presentation of our study results from the discussion of existing literature. In the revised manuscript, the Results section will focus solely on presenting the findings of our study in a clear and concise manner, using appropriate tables and figures to support the text. The Discussion section will then be dedicated to comparing and contrasting our results with existing literature.
Comment 5: What does "PDAC" stand for? Please make sure that all abbreviations are defined.
Response 5: We ensure that all abbreviations are defined upon first use in the manuscript. For "PDAC", we defined it as "Pancreatic Ductal Adenocarcinoma" in the Introduction section where it is first introduced. Additionally, we created a list of abbreviations and their definitions at the end of the manuscript for easy reference.

Reviewer 2 Report
Comments and Suggestions for Authors
This was a really interesting but difficult to read paper on using multiple bioinformatic tools to explore gene expression in paired and unpaired samples to derive a list of commonly (universally) differentially expressed genes. In PDAC or more widely PAAD, the use of these tools identified 19 genes that might represent biomarker or pharmacological genes of interest. The narrative through the paper is often quite difficult to understand and the description of some of the bioinformatics was rather obscure. I had to read the RankComp paper to understand what the authors had done in this study for example. The narrative of the resultsI found quite laboured as much of it is list-like with references to other studies where we know the genes of interest are functionally relevant in pancreatic cancer. Becuase of this, I was not 100% sure what teh ADVANCES were from this study. What have the authors discovered that we didn't previously know? I think this gets a little lost between the initial gene identification work and the drug:gene interaction work. Similarly, the description of the immune environment work is a difficult read: for example, the sentence "The results show that most up-regulated
UDEGs are significantly negatively correlated with the proportion of TME components
associated with response to ICB, including CD8+ T cells, naive B cell and activated
dendritic cells, and significantly positively correlated with the proportion of TME
components associated with resistance to ICB, including M0 macrophages, resting
dendritic cells and Tregs (Fig 5B)". I read this three of four times and still am not sure I understand it.
Part of the issue lies with the figures which I think need to be at much higher resoution - even on a review copy - for the reader to related p.18 to figure 5
I think the paper overall is very interesting and with some careful reworking and editing I would see it is publishable and of wider value.
Comments on the Quality of English Languagecommented on above
Author Response
We sincerely appreciate your detailed and constructive feedback on our paper. Your insights are invaluable in helping us improve the clarity and impact of our research.
1.Clarity of Narrative and Bioinformatics Description
We acknowledge that the narrative and bioinformatics descriptions in our paper were not as clear as they should be. To address this, we will rewrite the sections to make them more accessible. We will simplify complex language and provide more in - depth explanations of the bioinformatics tools and methods we used. For example, we will expand on the use of RankComp in a more step - by - step manner, without relying too much on readers referring to the original RankComp paper.
2.Defining the Advances of the Study
We understand your confusion regarding the advances of our study. In the oncogene and tumor suppressor gene section, we used the list from the CancerMine database to support our findings. We did this to build on prior research and strengthen the credibility of our gene-related results. Our key discovery is the identification of 19 genes in PDAC that have the potential to be novel biomarkers or pharmacological targets. These genes were found through a comprehensive analysis of paired and unpaired samples using bioinformatic tools, which is a unique approach in this context. While there are references to other studies showing the functional relevance of some of these genes, our study provides a new, integrated perspective. We also discovered relationships between the identified genes and the immune environment in pancreatic cancer, which has not been fully explored before. This new knowledge can potentially be translated into new treatment strategies, especially in the context of immunotherapy.
3.Clarifying the Immune Environment Results
We understand that the sentence regarding the immune environment was difficult to comprehend. We will rephrase it to improve clarity. For instance, we could break it down into smaller, more digestible sentences. We might rewrite it as follows: "Our results show relationships between up-regulated UDEGs and the components of the tumor microenvironment (TME). Specifically, most up-regulated UDEGs are significantly negatively correlated with the proportion of TME components related to a response to immune checkpoint blockade (ICB), such as CD8+ T cells, naive B cells, and activated dendritic cells. On the other hand, they are significantly positively correlated with the proportion of TME components associated with resistance to ICB, like M0 macrophages, resting dendritic cells, and Tregs (Fig 5B)."
We are committed to making these improvements to ensure that our paper is more understandable and that the significance of our research is clear. Thank you again for your time and effort in reviewing our work.
4.Figure resoution
Thank you very much for your astute observation regarding the resolution of the figures. We sincerely apologize for any inconvenience this might have caused to your review process.​
We fully understand the importance of high-resolution figures for a clear correlation between the content on page 18 and Figure 5. To address this issue, we have prepared tiff files with significantly higher resolution in
We hope that these new files will enhance the clarity of our manuscript and facilitate your review. Once again, thank you for your valuable feedback.
Round 2
Reviewer 1 Report
Comments and Suggestions for Authors
I would like to thank the authors for addressing my comments and significantly improving the manuscript. The manuscript is recommended for publication after addressing the following minor comments:
1) Fig 5 legend: Please correct TCGD "PADC" to "PDAC"
2)Appendix Figure 1: What is the "PAAD" cohort? I assume that it refers to "PDAC". If not, please update the abbreviations section of the manuscript.
Author Response
Comments 1. Fig 5 legend: Please correct TCGD "PADC" to "PDAC"
Response 1. We have meticulously located the relevant part in Fig 5 legend and have replaced “PADC” with “PDAC” as per your instruction. A thorough double - check has been carried out to ensure that this correction is consistent throughout the entire figure caption and related text in the manuscript.​
Comments 2) Appendix Figure 1: What is the "PAAD" cohort? I assume that it refers to "PDAC". If not, please update the abbreviations section of the manuscript.
Response 2. You are correct in your assumption. The "PAAD" cohort indeed refers to "PDAC". We apologize for the confusion caused by the inconsistent abbreviation. We have made corrections to all inconsistent parts in the text, including the horizontal axis in Appendix Figure 1. This update will make the manuscript more reader - friendly and avoid any potential misunderstandings in the future.
Reviewer 2 Report
Comments and Suggestions for Authors
The authors have improved this paper significantly, helping the non-expert bioinformatician to understand what they have done and what it means much more clearly than in the 1st submission.
I recommend publication
Author Response
We are delighted to hear that you recognize the substantial improvements in our paper. Our aim was to make the content more accessible to a wider audience, especially non - expert bioinformaticians. Your positive feedback validates our efforts, and we are committed to maintaining this level of clarity in all our future work.​
We firmly believe that these corrections and the overall improvements have further enhanced the quality and clarity of our manuscript. Thank you once again for your time, expertise, and support. We eagerly anticipate the successful publication of our manuscript.​